# Driving Deployment of Bioengineered Products—An Arduous, Sometimes Tedious, Challenging, Rewarding, Most Exciting Journey That Has to Be Made! [note 1]

**DOI:** 10.3390/bioengineering11080856

**Published:** 2024-08-22

**Authors:** Gordon George Wallace

**Affiliations:** Intelligent Polymer Research Institute, AIIM Facility, Innovation Campus, University of Wollongong, Squires Way, North Wollongong, NSW 2500, Australia; gwallace@uow.edu.au

**Keywords:** clinical translational, bioengineered products, neural regeneration, cartilage, islet cells, cornea

## Abstract

More than three decades ago, we embarked on a number of bioengineering explorations using the most advanced materials and fabrication methods. In every area we ventured into, it was our intention to ensure fundamental discoveries were deployed into the clinic to benefit patients. When we embarked on this journey, we did so without a road map, not even a compass, and so the path was arduous, sometimes tedious. Now, we can see the doorway to deployment on the near horizon. We now appreciate that overcoming the challenges has made this a rewarding and exciting journey. However, maybe we could have been here a lot sooner, and so maybe the lessons we have learned could benefit others and accelerate progress in clinical translation. Through a number of case studies, including neural regeneration, cartilage regeneration, skin regeneration, the 3D printing of capsules for islet cell transplantation, and the bioengineered cornea, here, we retrace our steps. We will summarise the journey to date, point out the obstacles encountered, and celebrate the translational impact. Then, we will provide a framework for project design with the clinical deployment of bioengineered products as the goal.

## 1. Introduction

The Intelligent Polymer Research Institute (IPRI) was established in 1990 by a visionary Vice Chancellor (VC) and the DVC Research group, who could see beyond the convenience of a conventional organisational structure [1]. There was a need to build capabilities outside conventional faculties that could work with those faculties to go above and beyond the usual focus on academic prowess. This high-level support was critical and became infectious. It permeated through the organisation, and the need to respond with agility and timeliness on interdisciplinary research matters underpinned our present and future success.

Having established IPRI and built national and international research networks, we were well placed to compete for the pinnacle in Australian Research (ARC) Funding—an Australian Research Council Centre of Excellence in 2005—where the ARC Centre for Electromaterials Science (ACES) emerged [2].

There are too many living individuals who contributed to that amazing venture to name them all here; however, you will find them listed as coauthors in my full publication list [3]. Let me name a few colleagues that are no longer with us. Prof Leon Kane Maguire—a legendary scientist and amazing individual who inspired connectivity and collegiality. Prof Alan MacDiarmid—another legend who inspired us all, who was chair of our International Advisory Board and totally embraced the Intelligent Polymers concept. Prof Don Iverson, who helped forge our medical technologies work. Dr Jeffrey Gin, who was fatally wounded while doing casual work during his PhD. Dr Peter Teasdale and Dr Peter Riley, IPRI graduates, also took too soon. Ultimately, the research enterprise and its clinical translational impact are about people, the relationships forged through debate and discussion, and our shared commitment to the collective vision. The determination that all should and will benefit from. The following specific case reports will serve to place the forgoing words in a richer context.

## 2. Case Studies in Bioengineering Translation

### 2.1. Neural Regeneration

Inherently conducting polymers [4] remain one of the most versatile and poorly understood groups of electromaterials on planet Earth. The inextricable/interdependent multifunctional behaviour that makes them impossible to fully understand using today’s approaches is almost certainly at the root of the extraordinary behaviour often observed. This includes the ability to act as a stimuli-responsive polymer in biosensors, with the earliest comprehensive description of such behaviour being found in our collaborative publication [5].

An example of unusual behaviour is found in the biosensor response based on high-affinity antibody–antigen binding being rendered reversible if pulsed potentials of a particular frequency are used [6]. Furthermore, the ability to incorporate living cells into the electroactive polymer at the time and site of synthesis and to extract sensory information as antibodies bind to those cells rendered these materials noteworthy [7]. It was these findings, along with our early work on the use of these materials to stimulate nerve cell growth, that attracted the attention of Prof. Graeme Clark (of BionicEar fame), and together we mapped a way to develop platforms to enhance neural communications. This resulted in a number of significant developments at the bench, where we demonstrated the ability to dramatically promote neurite outgrowth [8] as well as control the direction of such growth [9] (Figure 1).

In parallel studies, we discovered that we could control the development of muscle cells in a similar manner [10,11], with ICP-supported electrical stimulation promoting cell proliferation, myotube formation, and the development of muscle fibres. In other related studies, we encountered another inexplicable result using conducting polymer electrodes. We were able to reverse the deterioration of neural connections in cell cultures from a “schizophrenic mouse” [12]. Cells from Disc 1 and Neuregulin 1 (NRGI) knockout mice (Genes whose dysregulations are associated with neuronal diseases, including schizophrenia) formed the basis for this work. In collaborative work with Prof. Xu-Feng Huang (an expert on schizophrenia), it was verified that neurite outgrowth and connectivity were inhibited in cells obtained from knockout mice and cultured in vitro. However, following electrical stimulation through a conducting polymer, neurite outgrowth and connectivity were restored [13].

The electrical stimulation of organic conducting polymers can also be used to trigger the release of incorporated bioactive molecules such as proteins or drugs. We have demonstrated the use of electrorelease phenomena to investigate how the use of electrical energy generated by epileptic seizures could stimulate the release of an appropriate ameliorating drug [14]. A bench-top model was built to highlight the possibilities and further elucidate the challenges. It was revealed that the magnitude of the electrical impulses associated with seizures was insufficient in itself to stimulate drug release. However, future iterations of conducting polymers enabling signal amplification within an integrated structure are not beyond the realm of possibility.

We have recently tackled the challenge of developing wireless electrical stimulation protocols using conducting polymers. Wireless strategies will be critical for practical deployment. We have utilised “bipolar electrochemistry”, wherein reactions are driven by placing the redox substrate in an electric field. We have demonstrated the efficacy of this approach in communicating with neural cell lines [15,16,17] (Figure 2). This novel approach was introduced to our team by Prof. Robert Forster (DCU) over a coffee in Bath (the place in the UK, not the liquid container).

Given the extraordinary phenomena described above, we pursued the fabrication of conducting polymer structures using available tools, which led us to the development of novel printable inks [18]. We were able to fabricate useful structures containing organic conductors, and even printing living cells [19] was possible. Eventually, we embarked on the use of 3D printing [20] to create even more useful structures. Therefore, with all of these amazing technical findings, why could we not find a path to clinical deployment? What was the showstopper?

At the time of the developments described above, our ability to scale the processing of inherently conducting polymers in a form that could be fabricated into clinically relevant structures was limited. The inks we could produce for printing were limited in composition and bioactivity. Furthermore, the long-term stability of conducting polymers in vivo had not been demonstrated, so our initial target to replace platinum as the electrode material used in the bionic ear was going to be challenging. The inexplicable results, while tantalising to the scientific researcher, were unlikely to sit well with the regulators. In parallel, other organic conductors (carbon nanotubes and then graphene) emerged as having excellent potential as neural communication electrodes, and these materials were easier to process and fabricate into structures. Interestingly, our foray into graphene-based structures has brought us full circle—with graphene-based fibres providing unprecedented levels of fidelity in communication with nerve cells [21], underpinning advances in electroceuticals—in our collaborative work with Prof. Mario Romero-Ortega.

Graphene-containing structures also provided intriguing yet inexplicable outcomes when used in studies related to bone regeneration. Initial work involved the fabrication of 3D scaffolds compised of PDMS and graphene, demonstrating improved osteogenesis [22]. Subsequent studies indicated this could be further enhanced through electrical stimulation [23].

As with all technological developments targeted towards translation, the costs to traverse the path to deployment increase exponentially as we progress. Challenges for organic conductors include obtaining reliable source materials, scaling up, and the establishment of cGMP facilities for animal and possibly human studies. While addressable, these are costly and are only justified by market considerations.

On the regulatory side, the gold standard used for electrical stimulation, platinum, and IrOx, were already established and approved for implantation, but these would be difficult to replace. There may still be a place for organic conductors as new recording and stimulating electrodes in the clinic. However, a clear and compelling competitive advantage must be demonstrated. We are getting closer, but I fear the inability to control composition and structure rigorously and reproducibly will limit the use of components as regenerating implants.

### 2.2. Cartilage Regeneration

Our work on nerve and muscle repair, as well as our foray into the development of novel fabrication (printing) technologies, attracted the attention of Prof. Peter Choong, an orthopaedic surgeon based at St Vincent’s Hospital in Melbourne. As a surgeon often required to remove body components that were diseased, Peter is committed to using the most advanced technologies to restore as much function as possible. We made some progress in the use of artificial muscles based on conducting polymers [24] and demonstrated the ability to interface with real muscles [25]. However, for the reasons stated above, it was unlikely that conducting polymers would be deployed in the clinic at the time, but those studies catalysed many conversations, which is one of the hallmarks of translational science.

Over a regular 6:00 a.m. coffee in Melbourne, Peter suggested we use advanced fabrication and biomaterials to enhance a proposed stem cell therapy to regenerate cartilage. Oh, is that left field from what we were focussed on? The idea was to combine two advances: the use of stem cells to regenerate cartilage and the use of biomaterials, such as chitosan, as a tissue-engineered construct to support regeneration. Could we use one to improve the efficacy of the other? And here we go—initially using chitosan [26] and then Gelatin Methacryloyl (GelMA)/Hyaluronic Acid Methacryloyl (HAMA) [27] as the biomaterial platform—we forged ahead. The GelMA-HAMA constructs were achieved using a core-shell (co-axial) printing approach with customised delivery hardware designed and fabricated in our laboratories [28]. The use of Adipose stem cells from the infrapatellar fat pad located behind the knee proved efficacious [26]. However, our initial studies indicated that the retrieval and workup of these tissues and the cell extraction and proliferation steps required many days. Thinking of deployment in the clinic, we asked how we achieve this retrieval and cell processing steps in the time allowed for a single surgery: 6 h retrieval and processing challenges [29]. After much effort, work in this area is progressing, and the target is still in sight.

In positioning for translation, a grant funding opportunity under the Medical Research Future Fund (MRFF) emerged. This enabled us to integrate all of the skills, science, engineering, clinicians, regulatory, consumer advocacy groups, and commercial skills needed for translation. It is crucial that all stakeholders be involved early in the translational process. The path forward is now clear. Now we have a map and a compass for this one! Funding sources such as the MRFF are vital for translation.

Building on the cartilage regeneration in the knee adventure, we were approached by Prof. Payal Mukherjee, an ENT surgeon from RPA and COBLH. Payal is passionate about creating a solution to treat children with microtia using 3D biofabrication to create a living ear. This challenge is next level compared to cartilage regeneration in the knee. Now, we need to create a 3D structure that will retain shape and integrity while the cartilage develops. This requires a structural and gel–cell component to achieve our targeted outcomes. We have developed a hybrid printing approach [30] to achieve this and now our focus is on identifying the best source of cells and the optimisation of the cell retrieval–proliferation processes [31].

Being aware of the time travel required to reach clinical deployment of a bioengineered product, Payal introduced us to a colleague, Sophie Fleming, an amazing prosthetist based in Australia—one of only three. Can you believe that? We have developed a customised printer to create a 3D-printed silicone ear [32]. While this printer does not provide an adequate finish for a completed product, it can dramatically decrease the workflow time for the prosthetist who provides the final finish. We have investigated the use of a mobile phone to carry out the scanning necessary to make this prosthetic technology entirely deployable [33].

### 2.3. Skin Regeneration

Skin is an extraordinary organ, structurally and functionally. Just ask Prof. Fiona Wood, and you will be thoroughly engaged. The fascination will never wane. Given the amazing job that skin does in containing and protecting our organs, we have much to be thankful for in this amazing structure (Figure 3).

We have used a printing approach that allows for the 3D biofabrication of a structure containing a robust composition to induce mechanical support while also having a more cell-friendly gel environment in the interstitial volume. We demonstrated that this structure was capable of supporting the formation of the dermis and epidermis [34] layers of skin (Figure 4). Exquisitely engineered, the gel component was sacrificial, providing space within the structure for cell migration as well as the exchange of nutrients and waste products.

In other work, in collaboration with seaweed producers Venus Shell Systems, we have found that a biomolecule extracted from green algae (ulvan) has extraordinary properties when it comes to skin regeneration [35]. The addition of ulvan to a GelMa-based formulation has the extraordinary effect of improving both mechanical and biological function. Usually, one is achieved at the cost of the other. We are not near the end in terms of the ideal bioprinted structure to facilitate skin regeneration; challenges such as innervation [36] and hair regrowth remain.

Let me take a side step; in a separate study with Japanese collaborators, we investigated the effect of electrical stimulation using conducting polymers on hair growth, and we obtained some surprising results [37]. We found that electrical stimulation of human dermal papilla cells using organic conducting polymer electrodes as a preconditioning step prior to implantation improved hair regeneration (Figure 5).

With our skin in the game, our mission began in earnest through a chance encounter with Prof. Fiona Wood at a biofabrication conference in Wurzburg in 2018. Our collaboration with Fiona and her team has focussed on our translation and deployment mission [38].

We are learning so much on the journey. We are currently in pre-clinical studies and are very much guided by what is needed for clinical deployment. We are learning by participating in the surgery. There is no doubt that the focus on surgery emphasises the criteria that must be met for translation (Figure 6).

If you want to compare biomaterials and cell combinations, performance alone is not sufficient. Sourcing and clinical deployment through the existing regulatory system must be considered. If we want to compare delivery systems (e.g., bench top vs. handheld), they should be compared in a surgical environment.

### 2.4. The Bioengineered Cornea

The Intelligent Polymer Research Institute was contacted by a surgeon who had the same name as one of my previous Vice Chancellors at the University of Wollongong. It turns out that he was the Vice Chancellor’s son. Prof. Gerard Sutton came to us with the challenge of creating a bioengineered cornea, and together with Danielle Fisher (Organ Tissue Donation Service, Kogarah, NSW, Australia), we set about building the team with the myriad of expertise required to do just that.

The cornea is a fascinating biological structure (Figure 7). It is comprised of endothelial, stromal, and epithelial layers. If we zoom in on the stromal layer, we soon appreciate the challenges we face in creating a bioengineered structure to recapitulate its form and function. It is comprised of many layers of collagen, wherein each layer of collagen fibrils is presented in an aligned form. Each alternate layer presents that alignment in an orthogonal fashion. Think about that—how on earth did this meso-structure evolve?

Using a technique known as electrocompaction, we can assemble collagen films in an electric field. The field induces the coagulation of collagen molecules from the solution and can be used to direct the alignment and assembly of collagen fibrils. This fabrication approach has proven useful in creating both the stromal [39] and epithelial layers [40]. We have recently demonstrated the ability to assemble the complete cornea on the bench with cells within the endothelial, stromal, and epithelial layers, all intact.

Of course, many challenges remain in terms of the multidimensional aspects of this project: sourcing and processing the collagen, improving the efficacy of cell isolation, and processing and automating the fabrication procedure for the whole cornea. We have assembled the team to do this. We have been successful in attracting MRFF Frontier support, consolidating our efforts with colleagues from the Queensland University of Technology, the University of Melbourne, the Centre for Eye Research Australia, the University of Sydney, the Organ and Tissue Donation Service (NSW), and the University of Wollongong; with regulatory, commercial, scientific, engineering, and clinical expertise collaboratively in place, we can see the door to deployment [41].

## 3. Challenges and Opportunities

There are several emerging areas of research and technological developments that will underpin clinical deployment in the above areas, as well as other areas of bioengineering. These include but are not limited to

### 3.1. Sourcing of Materials

Whilst the discovery of new biomaterials remains an important component of bioengineering, the comprehensive evaluation of sustainably sourced and readily accepted materials is equally important [42]. Collagen is an excellent case in point. Animal-derived collagen has proven highly successful in bioengineering, even finding its way into some commercial products, such as Integra™ and Bicornate. The most commonly used collagens are land-animal-derived: porcine and bovine. However, using animal-derived collagen is not always acceptable due to biological differences with respect to implantation in humans, as well as cultural sensitivities.

Collagen derived from human tissue, such as skin (collagen I) or placenta (collagen IV), provides an attractive alternative. Our preliminary work on the extraction of collagen from fish skin is likewise promising, producing mechanically robust cytocompatible structures [43]. Our recent studies, however, show distinct differences in collagen derived from marine sources compared to porcine sources [44]. In particular, the thermal stability range is important, being much lower in marine sources and varying even between marine species.

The marine environment remains an untapped source of biomaterials with great potential. As mentioned previously [35], we are working closely with Venus Shell Systems to explore this area. Sources such as seaweed, mussels, sea slugs, and fish skin (e.g., mentioned above) provide a potentially rich source of biomaterials.

A reproducible and reliable source of materials is critical to any regulatory process. The ability to produce any material modification and formulation in a rigidly controlled environment acceptable for implementation in human trials is essential.

### 3.2. Lost in Translation

As we orient our fundamental discoveries toward clinical impact, we recognise several important advantages, limitations, and opportunities. In the following sections, we outline several of these.

#### 3.2.1. Contactless Characterisation

The quest to take bioengineering into the clinic challenges the appropriateness of conventional biological characterisation tools for finished products such as the cornea discussed above. In order to accurately analyse bioengineered structures and observe their dynamic behaviour over time, it is common practice to halt their growth at certain points in time and permanently remove them from their environment. Once removed, these structures are “fixed” in their current state and subject to a range of processing/preservation steps, often involving rapid dehydration, soaking in cytotoxic solutions, snap-freezing, and micron-level sectioning. Not only does this workflow limit our ability to reliably compare characteristics over time, but these harsh processing steps also inevitably change the in situ characteristics of the structures we propose to investigate. We have been interested in techniques such as microwaveconductivity [45] to determine electrical properties in a contactless manner and, more recently, ultrasound [46,47,48] to gather information on material and cellular distribution in 3D. Undoubtedly, this field of stand-off or contactless characterisation will continue to grow; it must if we are to provide quality control checks that do not destroy the sample and are acceptable to regulators.

#### 3.2.2. Predictive Analytics

The rich data to be gleaned from contactless characterisation at different points in the bioengineered product life cycle will be critical to providing the streamlined optimisation of “manufacturing processes”. Consider the manufacturing of the bioengineered cornea, wherein a myriad of processing parameters across collagen production, the cell processing of three different cell types, and finally, the biofabrication of each layer require optimisation. Traditional approaches would be time consuming and cumbersome. Perhaps machine learning algorithms can help?

To illustrate the power of this approach, we have partnered with Prof. Svetha Venkatesh and her team at Deakin University to demonstrate how the use of interactive machine-learning tools can significantly enhance the efficiency of optimisation in 3D-printing protocols [49], dramatically reducing the number of experiments required.

#### 3.2.3. Regulatory Frameworks

Regulatory authorities are grappling with the most effective way to introduce bioengineered products into the clinic [50]. Such products comprising biomaterials and cells do not fit neatly within existing regulatory frameworks—is it a device or a biologic? Furthermore, it is difficult to envisage a single framework for all bioengineered products: the protocols envisaged to treat damaged skin using the patient’s own cells, a process confined to the operating theatre, requires a different regulatory approach to that governing the manufacturing of a bioengineered cornea. It is so much easier to regulate mass-manufactured products such as metallic knee or hip implants, but now we must grapple with customised and highly personalised structures. Now, add to this regulatory environment the use of machine learning to optimise processes, and you can see why regulators and regulations are scrambling to keep up with technological advances. We are starting to see some of our trainees welcomed into their ranks, bringing perspectives to inform the future of practice. It is critical that clinicians, scientists, patient advocates, and engineers continue to engage in active, ongoing, trusted dialogue with regulators if patient benefits are to be maximised [51].

#### 3.2.4. Health Economics/Policy Making

If people do not pay for bioengineered products, then such products will not benefit the health system. Therefore, the argument must be made that investing in a new biomaterial-based approach will deliver outcomes that result in economic benefits. This may be in terms of fewer days in hospitals, fewer complications, fewer revisions, and more speedy recovery—outcomes weighed up against inputs such as the costs of materials and cell processing, delivery devices, and increased surgery times. However, it is incumbent upon the innovators to make the case for the “value proposition”.

For example, a collaborator has examined the health economics of islet cell transplantation in order to examine the financial sustainability relative to the traditional approach. Recognising the value of such analyses, we have joined forces with these collaborators to conduct a preliminary analysis to investigate the economic arguments around the use of 3D bioprinting to create scaffolds (the Trojan horse approach discussed above) to improve the efficacy of islet cell transplantation to treat Type 1 diabetes. This activity was integrated into the thesis of a PhD student working on the technical aspects of the project. There is no doubt that the questions raised and knowledge gained provided a most useful dimension to these studies.

### 3.3. A Translational Project Frameworks

There are a number of considerations when embracing a translational project framework. These are often presented as linear because that is how we write, not how we think. Indeed, they are highly iterative and fully integrative.

It is important to identify a compelling unmet health-related need and become an “expert” in that need area through primary and secondary research. Read, but principally, talk to people who know this stuff. This may take the form of clinical observations, individual interviews, panel discussions, and surveys. The goal here is to develop a comprehensive statement of need that is actionable. When passion-driven experts come forward to fulfil this need, as they have in IPRI, we create an environment for success. Can your technology address the actionable unmet need? If it cannot, and you are persuaded by the unmet need, then go in search of the appropriate technology. Manifest a prototype; it teaches you more about yourself than you care to learn. As you do, think about sourcing, scale-up, manufacturability, logistics, and delivery and regulatory challenges—is there a regulatory pathway? Can one be developed? Material and cell selection must take this question into account. Sterilisation, packaging, and transportation protocols will be important.

Conduct some preliminary health economics—if this technology works, will it make economic sense to deploy? Are there parallel outcomes along the way that might have a health economic benefit? Is there a commercial pathway? Product definitions may well change on the journey. This will be an iterative entanglement. Always have IP protection front of mind. Clearly and obviously, technical advances do not necessarily result in clinical deployment. They may be necessary but an insufficient condition for clinical success. The Stanford Biodesign framework offers an example of a pervasive and acknowledged approach [52]. However, I think we need a framework that also gives every study/publication a translational score—of course, that has a temporal dimension. Is it translational ready?

Would it not be good to think about these aspects up front, or is our current system—lots of food (knowledge) producers out there in the market—looking for the right chef and the best approach? Can we change decades of thinking around the value of knowledge creation versus utilisation? I think we can. I think we have to.

### 3.4. Facilitating Networks: e.g., Beyond Science

The journey above requires the development of integrated agile networks. Many of us have spent decades building such networks, and it may be one of the most valuable legacies we can pass on to the next generation of researchers, clinicians, regulators, and entrepreneurs. As part of our efforts to do this, I have partnered with Prof. Payal Mukherjee to found and develop the Beyond Science program within the Royal Australian College of Surgeons [53]. This program brings together next-generation clinicians, scientists, and engineers and facilitates connectivity to proposal projects along the translational pipeline.

## Figures and Tables

**Figure 1 bioengineering-11-00856-f001:**
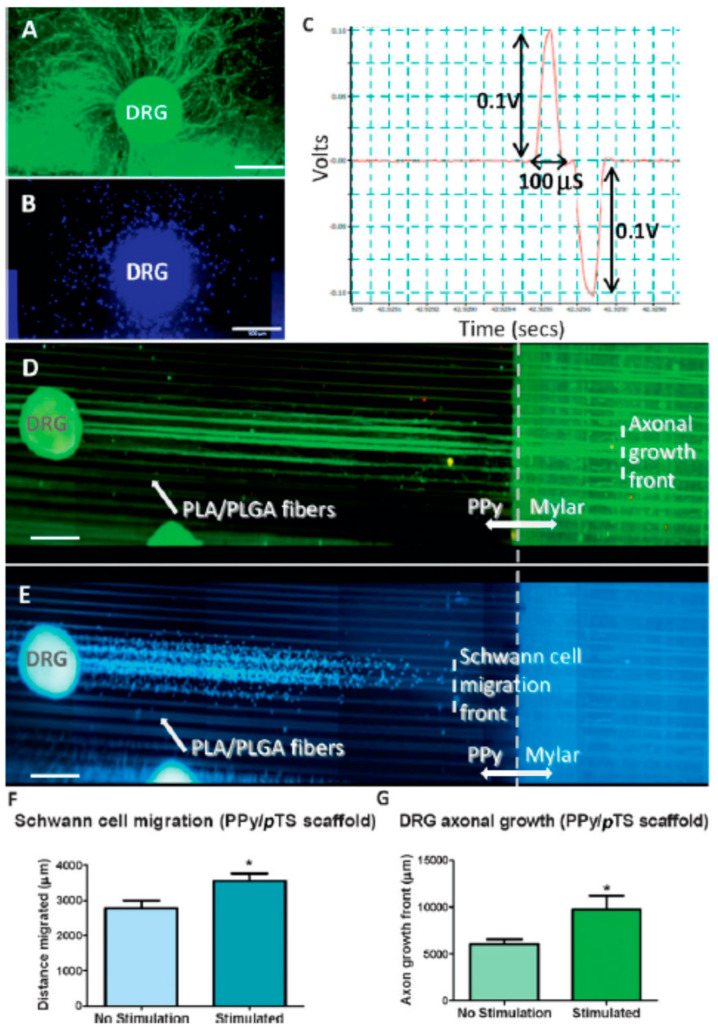
Dorsal Root Ganglia (DRG) were grown on chambers on a CAM-coated scaffold consisting of biodegradable fibres embedded in a PPy/pTS substrate, organised on a twin-electrode slide format to facilitate biphasic electrical stimulation of cells grown on the surface (**D**). DRG grown on plain PPy/pTS surfaces (without fibres) demonstrate radial axonal growth (**A**) and migration of Schwann cells (**B**) on stimulated and unstimulated polymers. In contrast, DRG explants grown and stimulated on hybrid PPy/pTS-PLA:PLGA (75:25) wet-spun polymer fibre scaffolds exhibited axonal growth (green) and Schwann cell migration (DAPI-labeled cell nuclei) in alignment with the wet-spun fibres (**E**,**F**). DRG were stimulated on the scaffold using an Accupulser system for 8 h per day for 4 days. The paradigm used for stimulation of DRG on the polymer scaffolds consisted of a bipolar pulse waveform with a 1 mA amplitude, biphasic pulses at 250 Hz with a 100 µs pulse width, a 3.78 ms interphase gap, and a 0.2 µs short circuit (**C**). Electrical stimulation significantly increased the extent of Schwann cell migration (**F**) as well as axonal growth (**G**). Scale bars ¼ 500 µm. (Reprinted with permission from Ref. [9], 2024, Wiley © 2008 WILEY-VCH Verlag GmbH & Co. KGaA).

**Figure 2 bioengineering-11-00856-f002:**
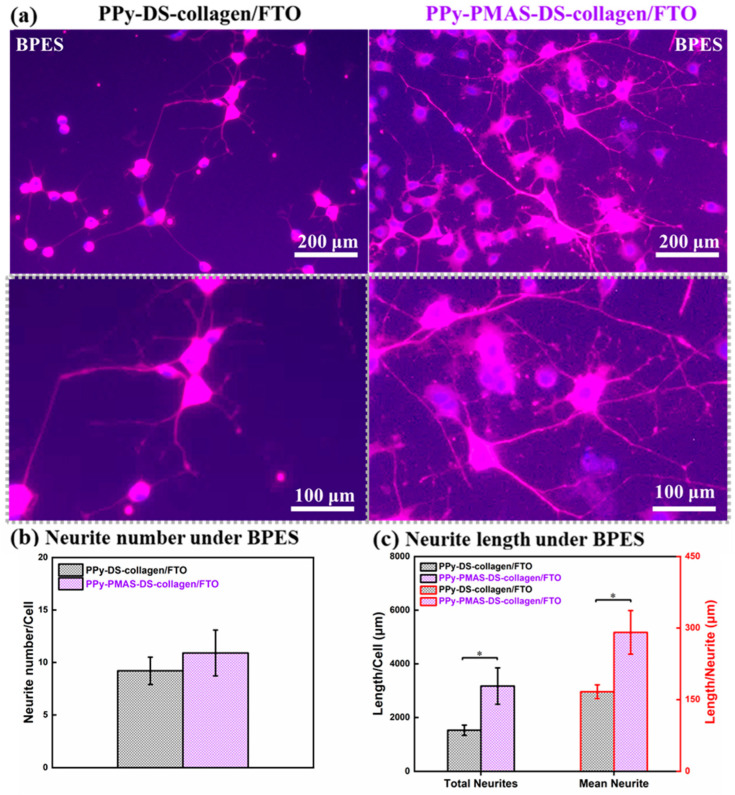
(**a**) Immunofluorescent images of PC 12 cells on day 7 after BPES treatment. (**b**,**c**) Assessments of neurite number and neurite growth of cells cultured on PPy-DS-collagen matrixes with or without PMAS modification, including (**b**) the total neurite number per cell, (**c**) the total length of neurites per cell, mean neurite length. BPES: cells were stimulated from day 2 for 8 h per day in continuous three days, then applied with standard culture for another three days. Statistical analysis used one-way ANOVA. Data are represented as mean ± standard deviation (SD) and “∗” (*p* < 0.05) was used to indicate significance (Obtained from [16]).

**Figure 3 bioengineering-11-00856-f003:**
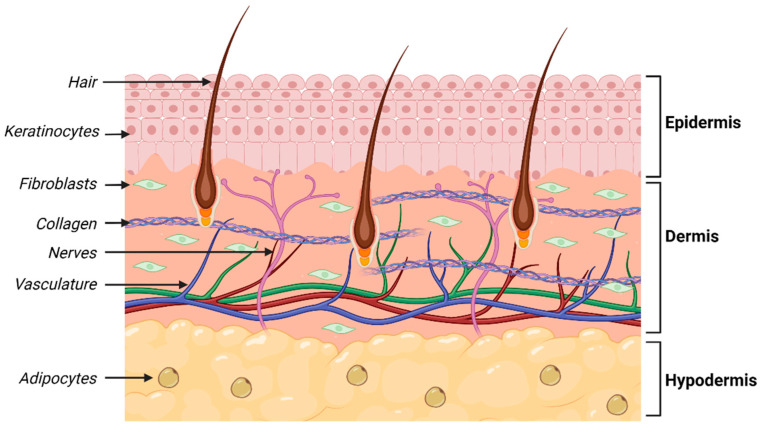
Schematic illustrating structure of human skin.

**Figure 4 bioengineering-11-00856-f004:**
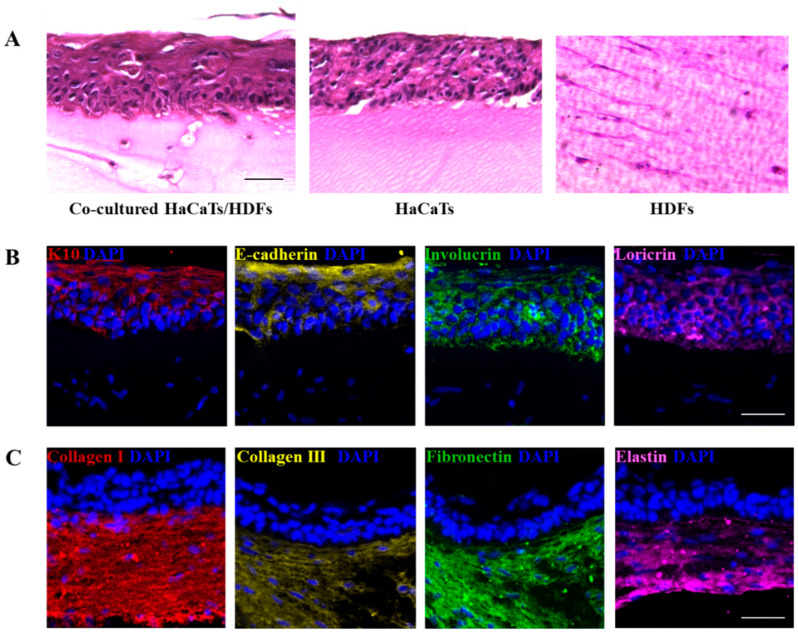
(**A**) Histological images of the fabricated double-layered skin-like constructs after 14 days culture. (**B**) K10, E-cadherin, involucrin and loricrin stained epidermis of the double-layered constructs after 14 days culture. (**C**) Collagen I, collagen III, fibronectin and elastin stained dermis of the double-layered constructs after 14 days culture. (Scale bars = 50 µm). (Reprinted with permission from Ref. [34], 2024, IOP Publishing).

**Figure 5 bioengineering-11-00856-f005:**
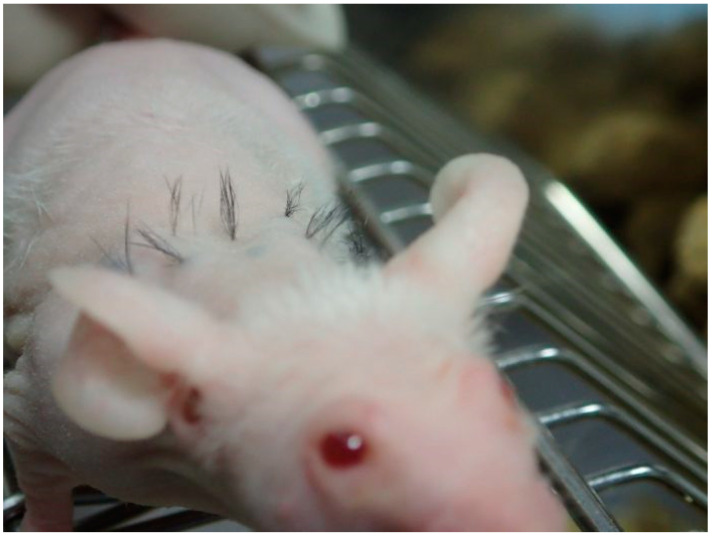
Hair regeneration on a mouse model. Photograph courtesy Fukuda Lab, Japan.

**Figure 6 bioengineering-11-00856-f006:**
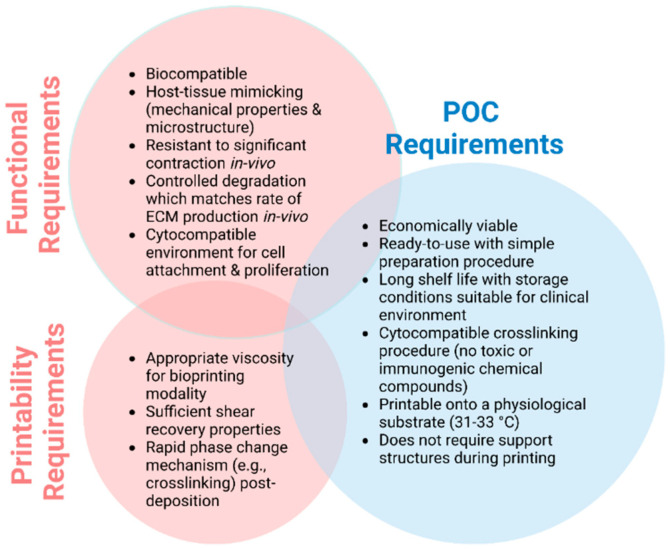
Ideal biomaterial requirements for use as an ink formulation at POC (Created with BioRender.com).

**Figure 7 bioengineering-11-00856-f007:**
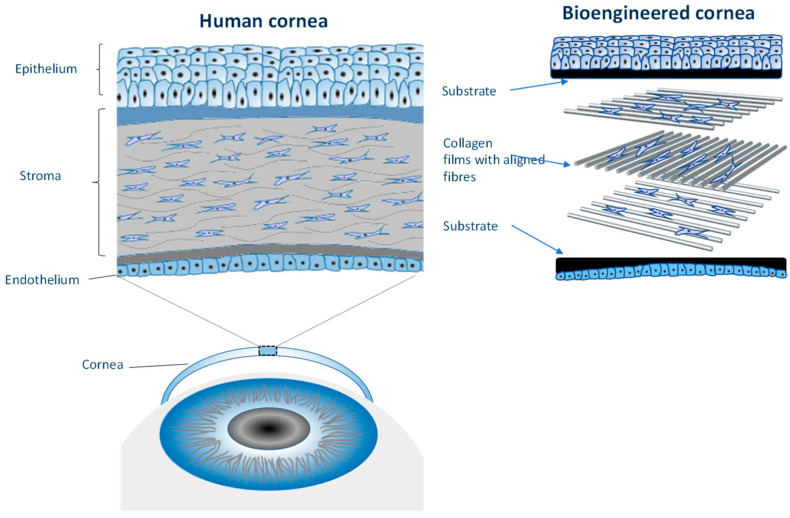
Schematic illustrating the structure of the cornea (Supplied by Dr Chen, IPRI).

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
