# Peer review of "Driving Deployment of Bioengineered Products—An Arduous, Sometimes Tedious, Challenging, Rewarding, Most Exciting Journey That Has to Be Made!†"

_bioengineering, 2024, doi:10.3390/bioengineering11080856_

Round 1

Reviewer 1 Report

Comments and Suggestions for Authors

This manuscript presents an extensive overview of the past three decades of bioengineering explorations undertaken by the authors. It highlights the journey from fundamental discoveries to clinical applications, highlighting key projects in neural regeneration, cartilage regeneration, skin regeneration, 3D printing, and the bioengineered cornea. The manuscript goals to share valuable lessons learned and provide a framework to expedite the clinical translation of bioengineered products. This manuscript provides more in-depth analysis and specific examples in each case study to illustrate the challenges and successes more effectively.

This manuscript may be accepted for publication.

Author Response

Thank you very much for taking the time to review this manuscript

Reviewer 2 Report

Comments and Suggestions for Authors

remove lines 63-67.

edit word spacing in line 150

line 159, edit end of sentence to remove )

Possible to expand more on the hair regeneration story?

Be sure to define all abbreviations used, examples in line 292

Thank you for an informative, educational, yet fun read!

Author Response

Thank you very much for taking the time to review this manuscript. Following Revierer's 2 suggestions, the manuscript has been accordingly revised.

Reviewer 3 Report

Comments and Suggestions for Authors

The work by Prof Wallace has been prepared in occasion of the 10th anniversary of Bioengineering, and then submitted to a collection edited by Prof Giuseppi-Elie to celebrate this important anniversary for the journal. In the spirit of what required by such a kind of collections, the Author offers a journey describing his own action in biotechnologies, an action that has been supported by collaborative cooperation among many groups involved in this field. In addition, he also provides a vision of what already done in his country and what to be done by the whole research community to incisively pursue important results in biotechnology research. Of course, all this by combining the efforts of researchers and respective institutions to set up the right strategies... strategies targeted to speed up the research action while retaining the high impact of as-found results. The Author has a massive track record of published works, for this reason his story-telling covers different aspects of applied Bioengineering. As it emerges since the first lines, the description of facts is ‘personal’, so I don’t feel to make comments only based on my own taste. However, I think that the reading of this work may also offer cues to researchers who are willing or are just approaching the research in biotechnology.  For this reason, and also taking into account what my mentor used to say (a pioneering vison is what favors the setting up of a launching pad towards the fulfilment of established goals... all this, upon a collective action), I suggest to publish the work as it is and I hope to see other works like this in the celebrative collection.

Author Response

(The authors gave the same response as above.)
